# Retinoic acid signaling mediates peripheral cone photoreceptor survival in a mouse model of retina degeneration

**Ryoji Amamoto, Grace K Wallick, Constance L Cepko***

Department of Genetics and Ophthalmology, Howard Hughes Medical Institute, Blavatnik Institute, Harvard Medical School, Boston, United States

**Abstract** Retinitis Pigmentosa (RP) is a progressive, debilitating visual disorder caused by mutations in a diverse set of genes. In both humans with RP and mouse models of RP, rod photoreceptor dysfunction leads to loss of night vision, and is followed by secondary cone photoreceptor dysfunction and degeneration, leading to loss of daylight color vision. A strategy to prevent secondary cone death could provide a general RP therapy to preserve daylight color vision regardless of the underlying mutation. In mouse models of RP, cones in the peripheral retina survive long-term, despite complete rod loss. The mechanism for such peripheral cone survival had not been explored. Here, we found that active retinoic acid (RA) signaling in peripheral Muller glia is necessary for the abnormally long survival of these peripheral cones. RA depletion by conditional knockout of RA synthesis enzymes, or overexpression of an RA degradation enzyme, abrogated the extended survival of peripheral cones. Conversely, constitutive activation of RA signaling in the central retina promoted long-term cone survival. These results indicate that RA signaling mediates the prolonged peripheral cone survival in the rd1 mouse model of retinal degeneration, and provide a basis for a generic strategy for cone survival in the many diseases that lead to loss of cone-mediated vision.

**\*For correspondence:**
cepko@genetics.med.harvard.edu

**Competing interest:** The authors declare that no competing interests exist.

## Editor's evaluation

Retinitis pigmentosa (RP) is a heterogeneous condition that leads to photoreceptor cell death and thus to different degree of blindness. The degeneration is often caused by mutations in genes expressed in rods but cones end up degenerating with a mutation-independent process. Thus, identifying how to save cones would increase the quality of life of RP patients, bypassing the tremendous genetic heterogeneity of the disease. This study shows that cones positioned in the periphery of the mouse retina is more resistant to cell death and investigates the reasons of this resilience. Using different genetic approaches, the authors elegantly show that retinoic acid signaling derived from Muller glial cells located in the periphery of the mouse retina is implicated in local survival of cone photoreceptors in mouse models of RP. This center-to-periphery asymmetry is also present in the human retina although the clinical significance of the observation remains to be tested. This well executed study will be of high interest to vision and ophthalmology science community

## Introduction

Retinitis Pigmentosa (RP) is a disease of the retina that affects 1 in 4000 people worldwide (*Hartong et al., 2006*). It is characterized by an initial loss of night vision, followed by loss of color and daylight vision. These symptoms are caused by the progressive degeneration of photoreceptors, rods and cones, which are necessary for achromatic night vision and daylight color vision, respectively. Although the disease progression is variable, a typical patient with RP loses night vision as an adolescent and

subsequently loses central daylight vision by the age of 60 (*Hartong et al., 2006*). Although early diagnosis is possible by procedures such as full-field retinal electroretinogram and dark adaptometry, there is no effective therapy for the vast majority of patients. Although gene therapy via gene augmentation has proven to be effective in humans (*Bainbridge et al., 2008*; *Cideciyan et al., 2008*; *Hauswirth et al., 2008*; *Maguire et al., 2008*), a one-by-one genetic augmentation approach is currently not feasible given that over 65 causative genes for RP have been identified (RetNet). However, a generic therapy aimed at preserving cone-mediated vision may be possible. Many RP patients have a disease gene that is expressed solely in rods, causing rod dysfunction and death (*Cideciyan et al., 1998*). Cones then suffer from bystander effects, leading to poor daylight and color vision, and in many cases, complete blindness.

The mechanisms underlying the non-cell autonomous, rod-dependent cone degeneration in RP are not completely understood. Previous studies have suggested several mechanisms to explain this secondary cone death, including toxicity from the degenerating rods, loss of rod-derived survival factors, oxidative damage, immune cell activation, necroptosis, and altered metabolism (*Ait-Ali et al., 2015*; *Guadagni et al., 2019*; *Gupta et al., 2003*; *Karlstetter et al., 2015*; *Léveillard et al., 2004*; *Murakami et al., 2012*; *Punzo et al., 2009*; *Ripps, 2002*; *Venkatesh et al., 2015*; *Viringipurampeer et al., 2019*; *Viringipurampeer et al., 2016*; *Wang et al., 2020*; *Xiong et al., 2015*; *Xue et al., 2021*). Ameliorating one or several of these mechanisms might then provide a generalized RP therapy to preserve daylight color vision. Intriguingly, we and others have noted that, in animal models of RP, the cones in the far periphery survive, even in the absence of surviving rods (*Cideciyan et al., 1998*; *Carter-Dawson et al., 1978*; *Charng et al., 2016*; *García-Ayuso et al., 2013*; *García-Fernández et al., 1995*; *Jiménez et al., 1996*; *LaVail et al., 1997*; *Lin et al., 2009*; *Ma et al., 1998*; *Milam et al., 1998*). The basis of this extended cone survival had not been addressed, though one hypothesis is that light exposure, which can induce photoreceptor degeneration (*Wenzel et al., 2005*), is reduced in the far periphery. However, while dark rearing can slow photoreceptor degeneration, particularly in albino RP animal models, it does not rescue the degeneration phenotype (*Chang et al., 2007*; *Fan et al., 2005*; *Naash, 1996*; *Pang et al., 2008*; *Paskowitz et al., 2006*). Additionally, cone survival in the far periphery is confined to an area demarcated by a sharp boundary, as opposed to a gradient, which would be expected if light exposure was the sole explanation. These results indicate that cone survival in the far periphery of the retina is not solely mediated by different levels of light exposure. Therefore, we asked whether there are molecular determinants that regulate cone survival in the far periphery, and if so, whether such determinants are sufficient to promote cone survival in the central retina.

We found that the regions of cone survival in the peripheral retina overlap with areas of active retinoic acid (RA) signaling. RA signaling regulates a wide range of activities such as cellular differentiation and survival in various organs including the central nervous system (*Cunningham and Duester, 2015*; *Maden, 2007*), but has not been reported to play a role in photoreceptor survival. In canonical RA signaling, RA is synthesized by Aldh1a1, Aldh1a2, or Aldh1a3, and it binds to RA receptors (RAR), which activate transcription of RA target genes via binding to genomic RA Response Elements (RARE) (*Cunningham and Duester, 2015*). During development of the mouse eye, *Aldh1a1* and *Aldh1a3* are expressed in the dorsal and ventral retinal periphery, respectively, with the expression of dorsal Aldh1a1 remaining into adulthood (*McCaffery et al., 1992*; *McCaffery et al., 1991*; *Sakai et al., 2004*). Deletion of the RA synthesis enzymes or the RA receptor in mice has revealed a role for paracrine RA signaling in the perioptic mesenchyme and choroidal vasculature, but not in retina patterning (*Cammas et al., 2010*; *Dupé et al., 2003*; *Fan, 2003*; *Goto et al., 2018*; *Molotkov et al., 2006*). Conditional knockout of all three synthesis enzymes in adult mice resulted in corneal thinning (*Kumar et al., 2017*). Through loss-of-function experiments, we found that RA produced from Muller glia (MG), the major non-neuronal cell type of the retina, is necessary for survival of the excess number of cones in the periphery. In addition, we found that constitutive RA signaling in MG is sufficient to promote cone survival in the more central retina. We also found that *ALDH1A1* expression is enriched in the human peripheral retina, suggesting that this mechanism of cone survival may play a role in human patients with RP.

# Results

## Cone photoreceptors survive in the peripheral retina where RA signaling is active

The rd1 mouse is a rapid photoreceptor degeneration model. It harbors a viral insertion and a point mutation in the *Pde6b* gene (*Han et al., 2013*), which is expressed only in rods. The rod death is considered a direct result of *Pde6b* loss-of-function (*Figure 1A*). Degeneration of rod photoreceptors starts at approximately postnatal day 11 (P11), and rapidly progresses, such that most rods degenerate by P20. As is the case with most RP cases in animal models and humans, cone photoreceptors then begin to lose function and degenerate (*Punzo et al., 2009*; *Milam et al., 1998*; *Figure 1A*). From the perspective of a retinal flatmount, both rods and cones degenerate in a center-to-periphery pattern. However, we and others (*Cideciyan et al., 1998*; *Carter-Dawson et al., 1978*; *Charng et al., 2016*; *García-Ayuso et al., 2013*; *García-Fernández et al., 1995*; *Jiménez et al., 1996*; *LaVail et al., 1997*; *Lin et al., 2009*; *Ma et al., 1998*; *Milam et al., 1998*) have noted that cones in the peripheral retina survive longer, although neither the extent nor the mechanism(s) of the extended peripheral cone survival had been explored.

To determine whether such peripheral cone survival persists long-term, we performed immunohistochemical (IHC) analysis of retinal flatmounts, which allows imaging of all cones in the retina. In agreement with previous studies, we found that Cone Arrestin[+] (CAR; also known as Arr3) cones were enriched in the peripheral retina at P40, with a preference towards the dorsal region (*Figure 1B*). CAR[+] cones in the dorsal periphery persisted at least up to P366, and interestingly, a nearly straight boundary demarcated the zone of survival (*Figure 1C*, left panel). Quantification of the number of cones in sampled peripheral and central regions showed that the peripheral retina contained significantly more cones compared to the central retina (*Figure 1C*). Such long-term peripheral cone survival was not restricted to the rd1 mouse model. The rd10 mouse model harbors a different mutation in the *Pde6b* gene, and the rate of photoreceptor degeneration is slower. Similarly, photoreceptor degeneration proceeds more slowly in the *Rho*[-/-] mouse model (*Rivas and Vecino, 2009*). In both rd10 and *Rho*[-/-] mouse models, cones survived long-term in the peripheral retina, with the same pattern as seen in rd1, persisting at least up to P366 and P531, respectively (*Figure 1C*). These results indicate that cones in the peripheral retina, especially dorsally, survive long-term in multiple models of rod-cone degeneration.

To understand the molecular mechanism that regulates cone survival in the periphery, we asked whether the peripheral cones were surviving because peripheral rods survived long-term. Previous ultrastructural examination of the rd1 retina revealed complete rod loss by P36 (*Carter-Dawson et al., 1978*). To corroborate, we performed single molecule fluorescent in situ hybridization (smFISH) for *Nrl*, a marker of rods, and *Arr3*, a marker of cones, in dorsal rd10 retinal sections at P366. While *Arr3*[+] cones were sparsely detected in the ONL, no *Nrl* signal was found (*Figure 1—figure supplement 1*). These results suggest that cones survive in the peripheral retina despite complete rod loss.

We reasoned that the peripheral cones might maintain an intrinsic transcriptional program that promotes cell survival and/or inhibits cell death. Central and peripheral cones from WT mouse retinas were thus profiled to determine if there were transcriptional differences that might underlie such intrinsic differences. To this end, an antibody-based FACS isolation strategy that we had previously developed for both fresh and frozen brain/retina tissue samples was used (*Amamoto et al., 2020*; *Molyneaux et al., 2015*; *Figure 1D*). Using the CAR antibody and FACS, a population of CAR[+] cells was easily identified and isolated, although the percentage of CAR[+] cells was higher than expected (Expected: ~ 3%, Observed: ~ 15%). Upon performing ddPCR on the extracted RNA, the FACS-isolated population was found to contain *Arr3*[+]/*Rxrg*[+] cones and *Glul*[+] MG, but not rods (*Figure 1—figure supplement 2*). Despite the contamination, SMART-Seq v4 cDNA libraries were generated and sequenced on NextSeq 500. The central and peripheral samples were analyzed for differential expression (DE). Among the DE genes, we found that *Aldh1a1* was highly enriched in the peripheral samples (*Figure 1E*), matching the previous data that it is expressed in dorsal MG (*McCaffery et al., 1992*). Functionally, Aldh1a1, Aldh1a2, and Aldh1a3 catalyze the synthesis of RA (RA), a metabolite that, when bound to RA receptor (RAR), drives transcription of RA target genes from genomic RA response elements (RAREs) (*Cunningham and Duester, 2015*; *Figure 1F*). To validate the expression pattern of *Aldh1a1*, we performed smFISH for *Aqp4*, a pan MG marker, and *Aldh1a1*, in P25 rd1 retinal sections. In the central retina, *Aldh1a1* signal was detected only at low levels. However, in the dorsal retina,

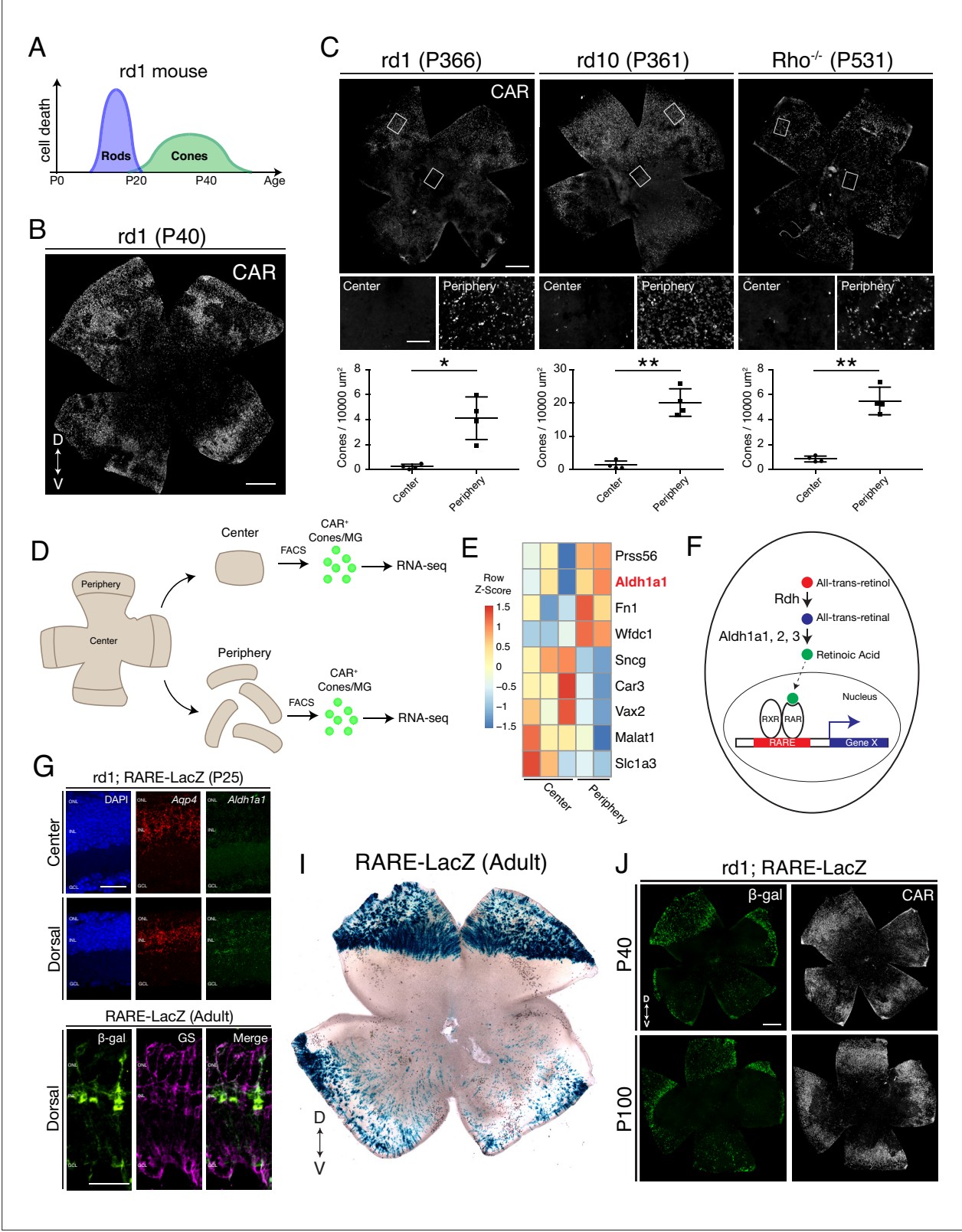

**Figure 1.** Cones in the rd1 mouse model survive long-term in regions with active RA signaling. (**A**) Schematic of the timeline of rod and cone photoreceptor death in the rd1 mouse model. (**B**) rd1 retinal flatmount IHC against Cone Arrestin (CAR) at P40. (**C**) Retinal flatmount IHC against CAR for rd1, rd10, and *Rho*$^{-/-}$ mouse models at P366, P361, and P531, respectively (top images). Center and peripheral insets used for cone quantification (middle images). Number of CAR$^+$ cones/10,000 um$^2$ (n = 4 each) in the center and periphery for rd1 (Student's two-tailed T test, p = 0.0206), rd10 (Student's

*Figure 1 continued on next page*

*Figure 1 continued*

two-tailed T test, p = 0.0036), and *Rho*$^{-/-}$ (Student's two-tailed T test, p = 0.0041). (**D**) Schematic of the strategy for cone-specific bulk RNA sequencing. Central and peripheral CD1 (non-degenerating strain) retinal tissues were collected, and CAR$^+$ cones (and contaminating MG) were FACS purified for downstream RNA sequencing. (**E**) A heatmap representing relative expression levels of differentially expressed genes (adjusted P-value < 0.05) between central and peripheral retina. (**F**) Simplified schematic of the RA signaling pathway. (**G**) SABER smFISH against *Aqp4*, a pan MG marker, and *Aldh1a1* in P25 rd1; RARE-LacZ dorsal and central retinal sections. (**H**) IHC against B-gal and GS, a MG marker, in adult RARE-LacZ retinal sections. (**I**) LacZ staining on adult RARE-LacZ flatmount. (**J**) IHC against B-gal and CAR in P40 (top row) and P100 (bottom row) rd1; RARE-LacZ flatmounts. D, Dorsal; V, Ventral; ONL, Outer Nuclear Layer; INL, Inner Nuclear Layer; GCL, Ganglion Cell Layer. Scale bars; 500 μm (**B, C, J**), 50 μm (**G, H**). All results are expressed as the Mean ± SD. *p < 0.05, **p < 0.01.

The online version of this article includes the following figure supplement(s) for figure 1:

**Figure supplement 1.** Long-term cone survival despite complete rod loss.

**Figure supplement 2.** Cone and MG enrichment in CAR$^+$ FACS-isolated cells.

**Figure supplement 3.** Aldh1a1 is expressed and RA signaling is active in peripheral MG.

**Figure supplement 4.** RA signaling activity is detected only in MG and a subset of ChAT$^+$ amacrine cells.

*Aldh1a1* puncta were distributed throughout the radial dimension of the retina, suggestive of an expression pattern in MG, as MG processes exhibit this widespread pattern (*Figure 1G*). The expression pattern of Aldh1a1 was also validated at the protein level (*Figure 1—figure supplement 3*).

A mouse strain, RARE-LacZ, can be used to identify transcriptional activation of the RA signaling pathway at a cellular level (*Rossant et al., 1991*). The read-out can be via IHC of the beta-galactosidase (B-gal) protein or chromogenic detection of its activity using X-gal. In sections, B-gal was mostly localized to cell bodies in the INL and processes spanning the retinal section, reminiscent of MG morphology (*Figure 1H*). Accordingly, B-gal staining overlapped with that of Glutamine Synthetase, an MG marker, indicating that most cells with activated RA signaling were MG, although not all peripheral MG were B-gal$^+$. B-gal was not expressed in any other cell type except ChAT$^+$ amacrine cells in the ganglion cell layer (*Figure 1—figure supplement 4*). Of note, regions of Aldh1a1 expression, and transcription from RA signaling, as indicated by B-gal expression, closely overlapped. This suggests that the RA synthesized by peripheral MG binds to receptors locally, and that the RA does not diffuse far enough to stimulate transcription activation in most nearby cells (*Figure 1—figure supplement 3*). In retinal flatmounts, the strongest B-gal activity was observed in the dorsal periphery, demarcated with a clear border, and weaker activity was found in the ventral periphery, as previously described (*Luo et al., 2004*; *Figure 1I*). To determine if this pattern of expression occurred in a retinal degeneration strain, the RARE-LacZ mice were crossed with rd1 mice. In rd1; RARE-lacZ mice, the spatial pattern of B-gal expression closely matched that of CAR$^+$ cone survival in rd1 mice, at both P40 and P100 (*Figure 1J*).

Taken together, these results indicate that cones in the rd1 mouse model survive long-term in regions which exhibit signaling via the RA receptor. These regions express the RA synthesizing enzyme, Aldh1a1, in MG, the cell type that exhibits active RA signaling.

## RA is necessary for survival of the excess peripheral cones

To determine whether RA is necessary for the longer cone survival seen in the periphery in a retinal degeneration strain, mice deficient in synthesis of RA in a retinal degeneration background were constructed. The rd1; RARE-LacZ mice were crossed to CAG-CreER and the *Aldh1a1*$^{fl/fl}$; *Aldh1a2*$^{fl/fl}$, *Aldh1a3*$^{fl/fl}$ mouse lines to generate a rd1; CAG-CreER; *Aldh1a1*$^{fl/fl}$; *Aldh1a2*$^{fl/fl}$, *Aldh1a3*$^{fl/fl}$ mouse line (henceforth referred to as Aldh Flox) (*Kumar et al., 2017*; *Figure 2A*). CAG-CreER was chosen as the Cre driver line to control the developmental stage at which the RA synthesis enzymes were knocked out, as RA is known to be a critical regulator of the development of many organ systems (*Cunningham and Duester, 2015*). The *Aldh1a2*$^{fl/fl}$ and *Aldh1a3*$^{fl/fl}$ alleles were included, despite the low expression levels of these genes in the adult retina, to minimize the potential effect of gene compensation. A single dose of tamoxifen was injected intraperitoneally at P10, a time point at which genesis of all retinal cell types is finished, and the retinas were harvested at P60 (*Figure 2A*). As assessed by ddPCR of the retina, the relative expression level of *Aldh1a1* was significantly reduced in retinas with CAG-CreER, compared to control littermates without CAG-CreER (Mean ± SD: -CreER: 0.01203 ± 0.005;+ CreER: 0.0003 ± 0.0002) (*Figure 2B*). In P60 Aldh flox retinal flatmounts without CAG-CreER, IHC for

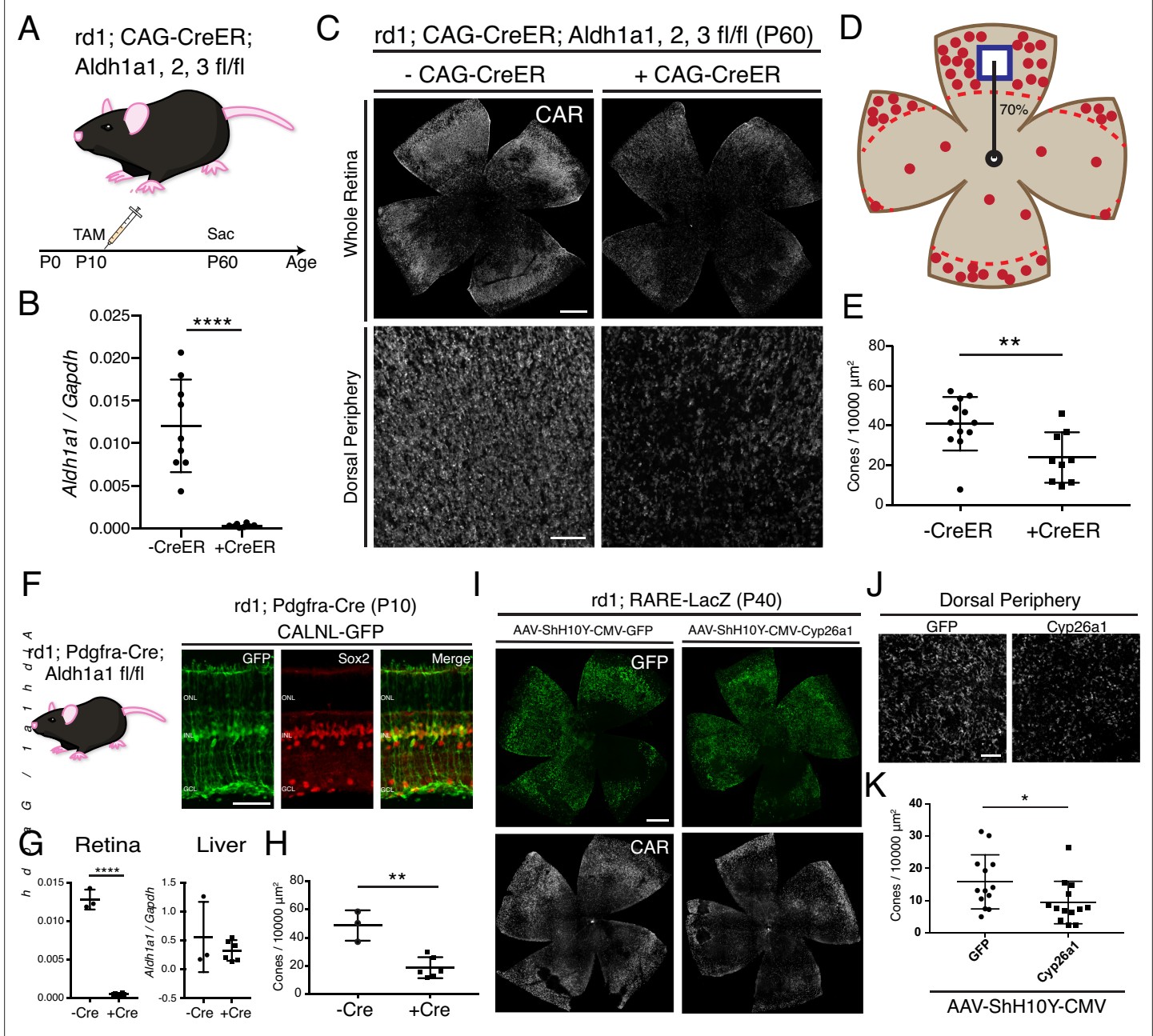

**Figure 2.** RA is necessary for peripheral cone survival. (**A**) Schematic of the experimental design. Aldh Flox mice were injected with tamoxifen to create a conditional knockout (cKO) of the RA synthetic enzymes at P10, and the retinas were harvested at P60. (**B**) Relative expression level of Aldh1a1 in retinas with or without CAG-CreER (-Cre: n = 9,+ Cre: n = 8, Student's two-tailed T test, p < 0.0001). (**C**) IHC against CAR in P60 Aldh Flox flatmounts showing the entire retina (top row) and the dorsal periphery (bottom row) with or without CAG-CreER. (**D**) Schematic of the area of cone quantification in the dorsal retina. Red dots represent surviving cones, and the blue box is the area of quantification. (**E**) Quantification of CAR+ cones in the sampled area of Aldh Flox retinas with or without CAG-CreER (-Cre: n = 12,+ Cre: n = 9, Student's two-tailed T test, p = 0.0086). (**F**) Validation of MG-specific Aldh1a1 cKO mouse line. IHC against GFP and Sox2 in P10 rd1; Pdgfra-Cre retinas electroporated with a Cre-dependent plasmid, CALNL-GFP. (**G**) Relative expression level of Aldh1a1 in retinas and liver with or without Pdgfra-Cre (Retina: -Cre: n = 3,+ Cre: n = 6, Student's two-tailed T test, p < 0.0001; Liver: -Cre: n = 3,+ Cre: n = 6, Student's two-tailed T test, p = 0.3893). (**H**) Quantification of CAR+ cones in the sampled area of Aldh Flox retinas with or without Pdgfra-Cre (-Cre: n = 3,+ Cre: n = 6, Student's two-tailed T test, p = 0.0016). (**I**) IHC against GFP (top row) and CAR (bottom row) in P40 rd1; RARE-LacZ flatmounts resulting from infection with AAV-ShH10Y-CMV-GFP (left column) or AAV-ShH10Y-CMV-Cyp26a1+ AAV-ShH10Y-CMV-GFP (right column). (**J**) Insets of the dorsal peripheral regions in both groups. (**K**) Quantification of CAR+ cones in the sampled area of infected retinas (GFP: n = 13, Cyp26a1: n = 13, Mann-Whitney test, p = 0.0441). ONL, Outer Nuclear Layer; INL, Inner Nuclear Layer; GCL, Ganglion Cell Layer. Scale bars; 500 μm (C top panels, **I**), 50 μm (**F**), 100 μm (C bottom panels, **J**). All results are expressed as the Mean ± SD. *p < 0.05, **p < 0.01, ****p < 0.0001.

*Figure 2 continued on next page*

*Figure 2 continued*

The online version of this article includes the following figure supplement(s) for figure 2:

**Figure supplement 1.** P4 subretinal injection of AAV-ShH10Y-CMV-GFP leads to retina-wide infection of MG and photoreceptors.

**Figure supplement 2.** *Cyp26a1* is overexpressed upon infection with AAV-ShH10Y-CMV-Cyp26a1.

CAR showed abundant cone survival in the periphery. In contrast, those with CAG-CreER appeared to have fewer cones in the periphery (*Figure 2C*). Accordingly, quantification of CAR$^+$ cones in sampled dorsal regions showed a significant reduction in the number of cones (Mean ± SD: -CreER: 40.94 ± 13.51;+ CreER: 23.9 ± 12.74), suggesting that RA is necessary for the locally enriched peripheral cone survival (*Figure 2E*). However, despite the near complete depletion of Aldh1a1 from the retina, some cones persisted in the dorsal periphery. Therefore, it is possible that other intrinsic or extrinsic pathways contribute to promote peripheral cone survival.

Due to the deletion of Aldh1a1, Aldh1a2, and Aldh1a3 in all cells resulting from the use of the broadly active CAG-CreER, it was possible that the effect of peripheral cone death was due to systemic toxicity. For example, hepatocytes in the liver express high levels of Aldh1a1 and regulate lipid metabolism (*Kiefer et al., 2012*). To overcome this issue, we generated another Aldh Flox mouse line with Pdgfra-Cre, which drives expression of Cre in MG (*Roesch et al., 2008*; *Figure 2F*). To validate the specificity of the Pdgfra-Cre, the DNA construct, CAG-LoxP-Neo-STOP-LoxP-GFP, which drives GFP expression in Cre$^+$ cells was electroporated into the retinas of rd1; Pdgfra-Cre mice at P0. At P10, GFP expression was mostly confined to MG, as evidenced by colocalization with Sox2, a marker of MG and a subset of amacrine cells (*Figure 2F*). Pdgfra-Cre mice were crossed with *Aldh1a1$^{fl/fl}$* mice to generate an MG-specific Aldh1a1 cKO mouse line. At P60, the retinas and the liver were harvested from littermates with and without Pdgfra-Cre. Using ddPCR, the number of transcripts for *Aldh1a1* was quantified from the retina and the liver. While *Aldh1a1* remained in the liver at high levels, it was essentially depleted in the retinas of mice with Pdgfra-Cre, indicating the specificity of the Cre line (Retina: Mean ± SD: -CreER: 0.0128 ± 0.001;+ CreER: 0.0005 ± 0.0001; Liver: Mean ± SD: -CreER: 0.5579 ± 0.6069;+ CreER: 0.3255 ± 0.1793) (*Figure 2G*). We then quantified the number of CAR$^+$ cones in the dorsal retina and found a significant reduction in the number of cones in the retinas with Pdgfra-Cre compared to controls (Mean ± SD: -CreER: 48.62 ± 10.77;+ CreER: 18.73 ± 7.44) (*Figure 2H*). These results indicate that MG-specific RA signaling is necessary for the excess cone survival in the periphery.

To further investigate whether RA is necessary for peripheral cone survival, another method was used to remove RA from the retina. Overexpression of Cyp26a1, an RA degrading enzyme, in the retina was carried out by AAV transduction. To maximally target MG, we used the ShH10Y capsid, which has a high tropism for retinal MG when injected subretinally or intravitreally (*Pellissier et al., 2014*). To obtain widespread AAV transduction in MG by subretinal injection, an injection time point was chosen such that most MG would have already been born (> P3), and the subretinal space would allow for a wide distribution of the vector. Subretinal injection with AAV-ShH10Y-CMV-GFP into P4 mice achieved high expression of GFP in central and peripheral MG, although some far peripheral MG were not transduced (*Figure 2—figure supplement 1*). Subretinal injection into the dorsal retina led to GFP expression throughout the dorsal retina, with partial infection of the ventral retina (*Figure 2I*). When combined with AAV-ShH10Y-CMV-Cyp26a1, smFISH showed expression of *Cyp26a1* in the dorsal retina (*Figure 2—figure supplement 2*). In accord with the cKO data, overexpression of Cyp26a1 significantly reduced the number of dorsal cones compared to GFP only controls (Mean ± SD: GFP: 15.81 ± 8.369;+ CreER: 9.416 ± 6.564) (*Figure 2J–K*).

Taken together, these results indicate that RA is necessary for the excess cone survival seen in the periphery of the rd1 mouse model.

## Activation of the RA signaling pathway promotes local cone survival in the central retina

To determine whether activation of the RA signaling pathway is sufficient for cone survival, the signaling pathway was activated in the central retina, independently of RA (*Figure 3A*). Canonically, the RARa-RXR dimer, bound to genomic RAREs, represses downstream transcription in the absence of the RA ligand (*Cunningham and Duester, 2015*; *Figure 3B*). In the presence of RA, the dimer recruits

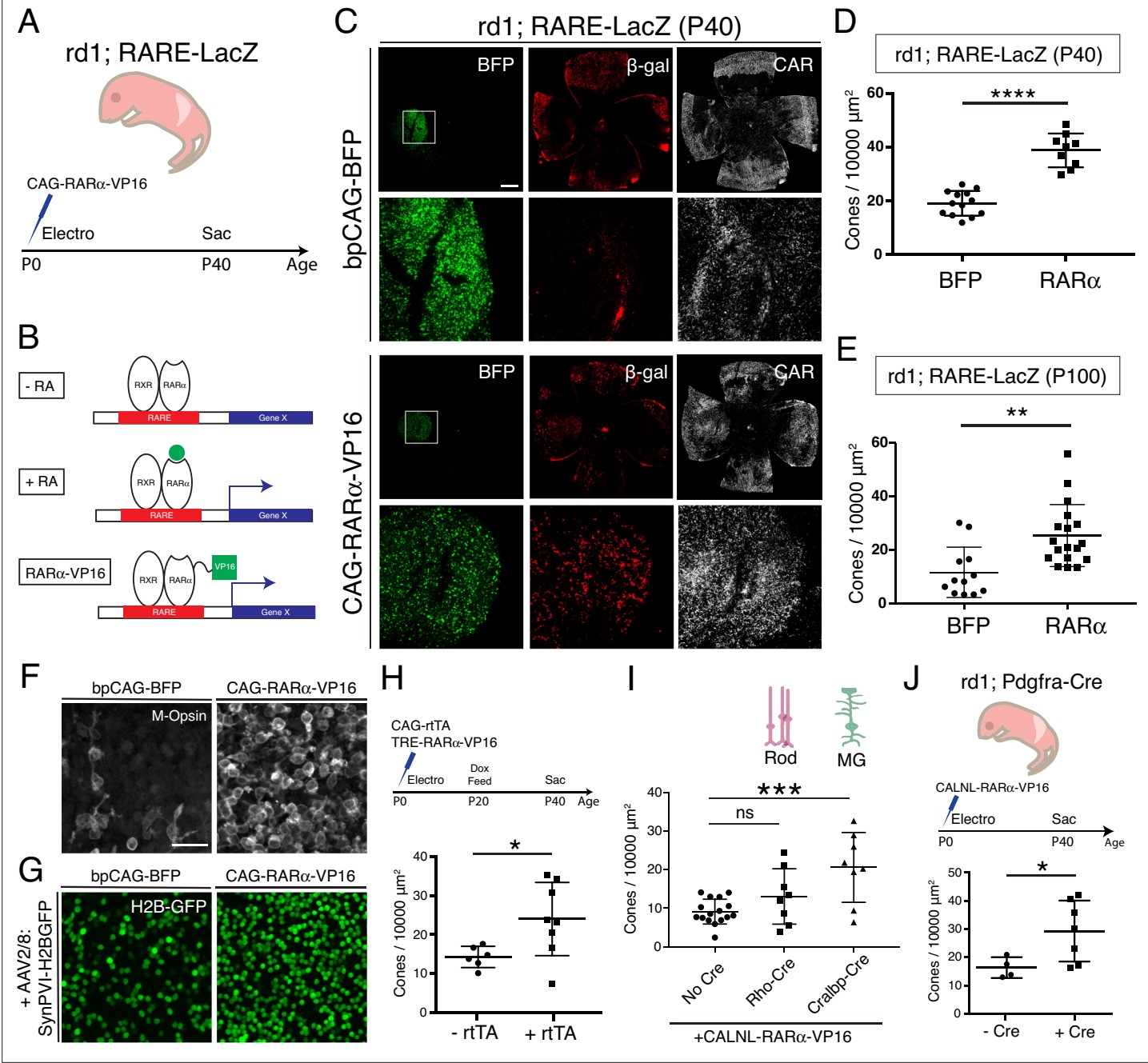

**Figure 3.** Constitutive RA signaling is sufficient for cone survival in the central retina. (**A**) Schematic of the experimental design. Retinas of rd1; RARE-LacZ mice were electroporated with bpCAG-BFP with or without the CAG-RARa-VP16 construct at P0 or P1 and harvested at P40. (**B**) Schematic of RA-mediated transcriptional activation and RARa-VP16-mediated constitutive activation. Without RA, the RARa-RXR dimer, bound to genomic RAREs, repress downstream transcription. With RA, the dimer activates transcription of downstream genes, which includes the RA degradation enzyme, Cyp26a1, creating a strong negative feedback. With the introduction of RARa-VP16, transcription of genes downstream of RAREs is activated despite the lack of RA. (**C**) IHC against BFP (left column), B-gal (middle column), and CAR (right column) in P40 rd1; RARE-LacZ whole retinal flatmounts (top rows) and BFP[+] insets (bottom rows) with overexpression of bpCAG-BFP with or without CAG-RARa-VP16. (**D**) Cone quantification in BFP[+] central retina of P40 rd1; RARE-LacZ mice electroporated with bpCAG-BFP with or without CAG-RARa-VP16 (BFP: n = 13, RARa: n = 9, Student's two-tailed T test, p < 0.0001). (**E**) Cone quantification in BFP[+] central retina of P100 rd1; RARE-LacZ mice electroporated with bpCAG-BFP with or without CAG-RARa-VP16 (BFP: n = 12, RARa: n = 18, Mann-Whitney test, p = 0.0010). (**F**) Representative images of IHC against M-Opsin (Opn1mw) in BFP[+] central retina of P40 rd1; RARE-LacZ mice electroporated with bpCAG-BFP with or without CAG-RARa-VP16. (**G**) Representative images of IHC against GFP in BFP[+] central retina of P40 rd1; RARE-LacZ mice electroporated with bpCAG-BFP with or without CAG-RARa-VP16 and injected with AAV8-SynPVI-H2BGFP. (**H**) Schematic of the experimental design (top). Retinas of rd1; RARE-LacZ mice were electroporated with TRE-RARa-VP16 with or without CAG-rtTA.

*Figure 3 continued on next page*

*Figure 3 continued*

Doxycycline was administered from P20 – P40, at which time point the retinas were harvested. Cone quantification in BFP+ central retina of P40 rd1; RARE-LacZ mice electroporated with with TRE-RARa-VP16 with or without CAG-rtTA (-rtTA: n = 6, RARa: n = 8, Student's two-tailed T test, p = 0.0311). (I) Cone quantification in BFP+ central retina of P40 rd1; RARE-LacZ mice electroporated with CALNL-RARa-VP16 with or without Rho-Cre or Cralbp-Cre (No Cre: n = 16, Rho: n = 8, Cralbp: n = 8, One-way ANOVA, *F* = 9.372, p = 0.0007; Tukey's multiple comparison test, No Cre vs. Rho: p = 0.3111, No Cre vs. Cralbp: p = 0.0005, Cralbp vs. Rho: p = 0.0511). (J) Schematic of the experimental design (top). Retinas of rd1; Pdgfra-Cre mice were electroporated with CALNL-RARa-VP16 and harvested at P40. Cone quantification in BFP+ central retina of P40 rd1; Pdgfra-Cre electroporated with CALNL-RARa-VP16 (-Cre: n = 4,+ Cre: n = 7, Student's two-tailed T test, p = 0.0488). Scale bars; 500 μm (**C**), 25 μm (**F**). All results are expressed as the Mean ± SD. *p < 0.05, **p < 0.01, ***p < 0.001, ****p < 0.0001.

The online version of this article includes the following figure supplement(s) for figure 3:

**Figure supplement 1.** *Cyp26a1* expression is upregulated upon RARa-VP16 overexpression.

**Figure supplement 2.** Cone quantification strategy in gain-of-function experiments.

co-transcriptional activators and the transcriptional machinery, leading to transcription of downstream genes. The RA signaling pathway is tightly regulated via a type of feedback inhibition, as the major downstream transcriptional target of RAR is Cyp26a1, which breaks down RA. To overcome this issue, a RARa-VP16 transactivator fusion protein was developed previously, to constitutively activate downstream transcriptional targets, even in the absence of RA (*Lipkin et al., 1996*; *Figure 3B*). A plasmid construct encoding this fusion, CAG-RARa-VP16, was electroporated into the ventral-central retina of rd1;RARE-LacZ mice with the electroporation control construct bpCAG-BFP. Retinas were harvested at P40 and cones were quantified in the electroporated patches (*Figure 3A*). In the bpCAG-BFP only control, the BFP+ electroporated patches were localized to the ventral-central retina, but B-gal expression was not induced, and CAR+ cones were sparsely distributed (*Figure 3C*). In contrast, when bpCAG-BFP was combined with CAG-RARa-VP16, a strong induction of B-gal and *Cyp26a1* expression was observed, indicative of transcription from RA signaling (*Figure 3C* and *Figure 3—figure supplement 1*). Within the electroporated patches that included CAG-RARa-VP16, a high density of cones was observed (*Figure 3C*). Quantification of cones in the electroporated region showed a significant increase in the RARa-VP16 group compared to controls (Mean ± SD: BFP: 19.14 ± 4.589; RARa: 38.8 ± 6.261) (*Figure 3D*, *Figure 3—figure supplement 2*). Of note, electroporation of the postnatal mouse retina results in expression of constructs only in late-born cell types of the retina, including rods, MG, bipolar cells, and amacrine cells, but not cones (*Matsuda and Cepko, 2004*). Therefore, this effect on cone survival is not due to expression of the constructs within cones, that is, it is a non-cell autonomous effect on cones. Furthermore, the region of increased cone survival was confined to the electroporated patch, with a sharp boundary, suggesting that the effect of RA signaling on cone survival is restricted locally.

To determine whether the cone survival effect of RA signaling is long lasting, cones were quantified at P100. Based on BFP, the expression level resulting from electroporation diminished over time between P40 and P100. Nevertheless, cone survival was significantly increased in the RARa-VP16 group compared to controls even at P100 (Mean ± SD: BFP: 11.67 ± 9.388; RARa: 25.41 ± 11.58), indicating that this effect lasts over an extended period of time. As RA signaling has been shown to increase the expression level of CAR itself, we sought to verify that RA signaling promoted bona fide cone survival, not just increased CAR expression, as described previously (*Wagner et al., 1997*). To this end, IHC for M-opsin (Opn1mw) was carried out, which showed that the surviving cones also expressed M-opsin (*Figure 3F*). To further verify the identity of these cones, during electroporation, AAV8-SynPVI-H2BGFP was co-delivered. This AAV encodes a construct which drives the expression of nuclear H2B-GFP from a synthetic Gnat2-based cone promoter (*Jüttner et al., 2019*). The surviving cones in the electroporated patch also expressed H2B-GFP, suggesting that these cones were positive for Gnat2, a cone marker (*Figure 3G*). These results show that RA signaling leads to survival of cones, instead of simply upregulating the expression of CAR.

Due to the experimental approach of constitutively activating RA signaling by overexpressing CAG-RARa-VP16 via electroporation at P0-P1, it remained unclear which cell type was responsible for the cone survival effect and when the effect was taking place. Electroporation of the retinas of newborn pups leads to expression within a few days in cell types born postnatally, including rods, MG, bipolar cells, and amacrine cells. Therefore, we sought to narrow the time period and cell type(s) that were sufficient for promoting cone survival by RA signaling. To determine the sufficient time

point, a Dox-inducible TetON system was used to control the timing of expression. The constructs, CAG-rtTA and TRE-RARa-VP16, were electroporated at P0-P1, and expression was induced from P20 – P40 by Dox administration. With such a paradigm, a significant increase in the number of cones in the rtTA+ group compared to controls was observed (Mean ± SD: -rtTA: 14.25 ± 2.723;+ rtTA: 24 ± 9.412), indicating that RA signaling during cone degeneration, and after rod degeneration (after P20), is sufficient for cone survival (*Figure 3H*). To determine which cell type was involved in promoting cone survival by RA signaling, specific promoters were used to drive expression of RARa-VP16. Rods are the most abundant of the electroporated cell types, while MG are the cell types that expresses Aldh1a1, and thus promoters for these two cell types were used (*Figure 1G*). Rho-Cre was used to activate expression in rods and Cralbp-Cre was used for expression in MG. Constructs with these promoters were co-electroporated with a Cre-dependent RARa-VP16 construct (CALNL-RARa-VP16). This strategy combined the specificity of the cell-type-specific promoters and the high expression level of the CAG promoter. Co-electroporation with Cralbp-Cre, but not Rho-Cre, showed significant increases in the number of cones compared to controls (No Cre) (Mean ± SD: No Cre: 9.125 ± 3.22; Rho: 13.07 ± 7.175; Cralbp: 20.61 ± 9.048) (*Figure 3I*). To further investigate the cell type question, the CALNL-RARa-VP16 construct was electroporated into rd1; Pdgfra-Cre retinas, which express Cre in MG (*Figure 2F*). Overexpression of RARa-VP16 in the Cre+ retinas increased the number of cones compared to controls without Cre (Mean ± SD: -Cre: 16.32 ± 3.681;+ Cre: 29.25 ± 10.79) (*Figure 3J*), in keeping with the results from Cralbp-Cre electroporation.

Taken together, these results indicate that RA signaling activation in MG during cone degeneration (> P20) is sufficient to promote cone survival in regions without endogenous RA synthesis.

## ALDH1A1 is enriched in the peripheral adult human retina

Whether RA signaling plays a role in preserving cones in human RP patients is not known. For a typical patient with RP, photoreceptor degeneration starts in the mid-peripheral retina, leading to loss of visual function in that region (*Milam et al., 1998*). Degeneration progresses over time towards the central retina, including the macula, with subsequent loss of central vision. However, peripheral vision is typically not assayed in humans, although islands of vision in the far periphery have been observed in some RP patients by functional visual testing (*Charng et al., 2016*; *Li et al., 1994*; *Patel et al.,*

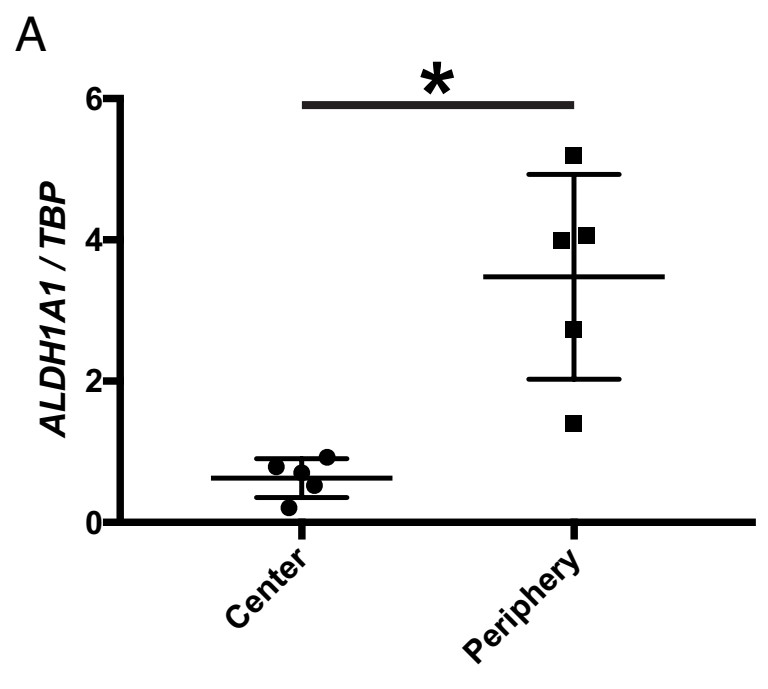

**Figure 4.** ALDH1A1 is enriched in the peripheral retina of humans. (**A**) Relative expression level of *ALDH1A1* compared to loading control, TBP, in the center and periphery of the human non-RP retinas (Center: n = 5, Periphery: n = 5, Two-way ANOVA, *P* = 0.0146). Figure Supplements.

*2022*). A systematic study of the geography of vision loss, which includes the far periphery, and the physical correlates, has not been done, but might yield insights into whether and how such islands may be preserved in humans. We obtained five postmortem adult human non-RP retinas to investigate the expression level of *ALDH1A1* in the central and peripheral retina. As assessed by ddPCR, *ALDH1A1* was significantly enriched in the peripheral retina (*Figure 4*). Furthermore, we compared our RNA-seq dataset with previously-published human transcriptional profiling of central and peripheral retina. In addition to the peripherally enriched *ALDH1A1*, *PRSS56* was enriched in the peripheral human retina while *SNCG* was enriched in the central human retina (*Orozco et al., 2020*; *Voigt et al., 2019*). This result opens the possibility that RA signaling is active in the human peripheral retina and may play a role in preserving peripheral vision, if any. In addition, other pathways which differ between peripheral and central retina may be conserved and might deserve investigation.

## Discussion

In this study, we set out to elucidate the molecular and cellular mechanism underlying long-term peripheral cone survival in the rd1 mouse model of retinal degeneration. In many mouse models of RP, cones in the far periphery survive long-term despite complete rod degeneration, as we also found here for rd1, rd10, and *Rho*⁻/⁻. We found that peripheral MG express Aldh1a1, leading to active transcription of RA target genes, and that the regions with active RA signaling closely matched those with surviving cones. Removal of RA, either by conditional knockout of the synthesis enzymes or by overexpression of the degradation enzyme, decreased the number of peripheral cones. Conversely, constitutive activation of the RA signaling pathway in MG promoted cone survival in the central retina. Taken together, RA signaling is both necessary and sufficient for the long-term survival of cones in the rd1 mouse model of retinal degeneration.

We found that peripheral MG express the RA-synthesizing enzyme, Aldh1a1, and activate transcription of RA target genes, autonomously or only within nearby MG. Importantly, RA signaling is not activated directly in cones; and therefore, the mechanism that promotes cone survival is non-cell autonomous with regards to cones. To gain a deeper mechanistic insight into this RA signaling-mediated cone survival, identification of the downstream effector is of great interest. To start, transcriptional profiling of MG and cones from Aldh Flox cKO mice and RARa-VP16 overexpression mice will be helpful to identify candidate ligand-receptor pairs that may play a role in promoting cone survival. However, given the diverse functions of MG, RA signaling may mediate a more complex effect on the tissue environment that secondarily promotes cone survival. For example, RA signaling plays a role in maintaining the integrity of the blood-retina-barrier (BRB) in the zebrafish eye, and BRB leakage is associated with cone degeneration (*Ivanova et al., 2019*; *Pollock et al., 2018*). Therefore, it is possible that RA signaling helps keep the BRB in the peripheral retina intact, which in turn helps cones survive, without directly influencing cones. Although the cause of secondary cone degeneration is currently unknown, use of RA signaling as a molecular handle may lead to a deeper mechanistic understanding. For practical applications as a mutation-agnostic gene therapy for patients with RP, identification of the downstream effector will be critical.

To go forward in the clinic, safety and efficacy are key components. For safety, discovery of an effector molecule, which may be less pleiotropic than RARa-VP16, may be important, as overexpression of a constitutively active transcription factor will lead to upregulation of myriad genes, potentially with deleterious effects. Gene therapy with an effector molecule also may be more effective than overexpression of RARa-VP16. Currently, AAV-based overexpression of RARa-VP16, even with a strong CMV promoter, was not sufficient to promote cone survival as the expression level was too low. Direct overexpression of an effector molecule(s), with cell type specificity, likely will be a more effective approach for gene therapy. Another reason to identify the downstream effector(s) is to avoid potential retinal ganglion cell (RGC) hyperactivity induced by RA signaling, as was recently reported (*Denlinger et al., 2020*; *Telias et al., 2019*; *Telias et al., 2021*). Although our current gain-of-function and loss-of-function approaches ensure that RGCs are not affected due to the specificity of the promoters and the injection method, it would still be a concern to promote cone survival at the cost of inducing detrimental RGC hyperactivity.

The human central retina is more susceptible to photoreceptor death compared to the peripheral retina in both RP and age-related macular degeneration (AMD) (*Charng et al., 2016*; *Fleckenstein et al., 2021*), and it remains unclear what molecular mechanisms underlie this difference. One

of the markers of the human central retina is the sustained expression of *CYP26A1* from embryo into adulthood (*Peng et al., 2019*; *da Silva and Cepko, 2017*). Furthermore, we have shown that *ALDH1A1* remains enriched in the adult human peripheral retina. While the function of these genes with opposing functions is unknown in the human retina, development of the high acuity area in the chick, which lacks rods as does the human fovea, requires inhibition of RA signaling (*da Silva and Cepko, 2017*). Although it remains unclear whether CYP26A1 in the human retina is required for foveal development, the lack of RA signaling in the central retina, perhaps as a remnant of development, may play a role in the vulnerability of the central retina to diseases such as AMD and RP. Conversely, activated RA signaling by ALDH1A1 may play a role in promoting cone survival in the far peripheral islands of vision seen in patients with RP (*Charng et al., 2016*; *Li et al., 1994*; *Patel et al., 2022*). To further support this hypothesis, it will be beneficial to correlate the expression pattern of *ALDH1A1* with regions of extended cone survival in people with RP.

## Ideas and speculation

It remains unclear why RA signaling is active in the peripheral adult retina. We speculate that Aldh1a1 expression in the dorsal periphery is a remnant of development, that is, its effect on cone survival is not due to a selective pressure for cone survival, but lack of a selective pressure to turn this gene off when its role in development is complete. During mouse and chick retinal development, *Aldh1a1* and *Aldh1a3* are expressed in retinal progenitor cells (RPC) of the dorsal and ventral peripheral regions, respectively (*McCaffery et al., 1992*; *Goto et al., 2018*; *Sen et al., 2005*). RA signaling in the chick retina plays a role in retinal patterning, as it is required for the correct cellular composition of the chick high acuity area, while in the mouse retina, it influences the perioptic mesenchyme and choroidal vasculature, but not retinal patterning (*Cammas et al., 2010*; *Dupé et al., 2003*; *Fan, 2003*; *Goto et al., 2018*; *Molotkov et al., 2006*). Postnatally, ventral *Aldh1a3* expression gradually disappears while dorsal *Aldh1a1* expression persists in adult MG. However, no overt retinal phenotype has been reported when Aldh1a1, 2, and 3 were conditionally deleted in the adult mouse, bringing into question whether RA signaling has any function in the adult retina. As adult MG have many of the molecular and morphological features of developing RPCs (*Jadhav et al., 2009*), we speculate that dorsal RPC-specific *Aldh1a1* expression persists into adult MG-specific expression as part of the dorsal peripheral RPC gene expression program. Interestingly, as the Aldh1a3 expression is turned off in the ventral MG in the adult, it might be the case that there is a selective pressure in the ventral region to eliminate expression of this enzyme after development is complete. As RA is an important regulator of vasculogenesis and the development of the BRB (*Pawlikowski et al., 2019*), and the main intra-retinal vasculature enters the retina via the ventral structure, the optic stalk (later the optic disc) (*Hyatt et al., 1996*), it might be the case that RA synthesis by ventral MG after development is somehow incompatible with the proper regulation of these activities.

## Materials and methods

### Key resources table

| Reagent type (species) or resource | Designation | Source or reference | Identifiers | Additional information |
|---|---|---|---|---|
| Genetic reagent (*Mus musculus*) | RARE-LacZ | Jackson Laboratories | 008477 | |
| Genetic reagent (*Mus musculus*) | CAG-CreER | Jackson Laboratories | 004682 | |
| Genetic reagent (*Mus musculus*) | Pdgfra-Cre | Jackson Laboratories | 013148 | |
| Genetic reagent (*Mus musculus*) | *Aldh1a1, 2, 3* Flox/Flox | PMCID: PMC5363406 | | |
| Recombinant DNA reagent | Rho-Cre | PMCID: PMC1764220 | | |
| Recombinant DNA reagent | Cralbp-Cre | PMCID: PMC1764220 | | |
| Recombinant DNA reagent | CAG-RARa-VP16 | This Study | | Expression of RARa-VP16 from CAG promoter |
| Recombinant DNA reagent | CALNL-RARa-VP16 | This Study | | Cre-dependent expression of RARa-VP16 from CAG promoter |
| Recombinant DNA reagent | TRE-RARa-VP16 | This Study | | Doxycycline-dependent expression of RARa-VP16 |

*Continued on next page*

*Continued*

| Reagent type (species) or resource | Designation | Source or reference | Identifiers | Additional information |
|---|---|---|---|---|
| Recombinant DNA reagent | CAG-rtTA | This Study | | Expression of rtTA from CAG promoter |
| Recombinant DNA reagent | AAV-CMV-Cyp26a1 | This Study | | AAV expression plasmid encoding Cyp26a1 from CMV promoter |
| Antibody | (Rabbit polyclonal) anti-Cone Arrestin | Millipore Sigma | AB15282 | (1:3000) |
| Antibody | (Chicken polyclonal) anti-B-galactosidase | Aves Labs | BGL1010 | (1:3000) |
| Antibody | (Goat polyclonal) anti-B-galactosidase | AbD Serotec | 4600–1409 | (1:3000) |
| Antibody | (Mouse monoclonal) anti-Brn3a | Millipore Sigma | MAB1585 | (1:500) |
| Antibody | (Rabbit polyclonal) anti-Iba1 | GeneTex | GTX100042 | (1:500) |
| Antibody | (Mouse monoclonal) anti-Glutamine Synthetase | Millipore Sigma | MAB302 | (1:3000) |
| Antibody | (Goat polyclonal) anti-ChAT | Millipore Sigma | AB144P | (1:500) |
| Antibody | (Chicken polyclonal) anti-GFP | Aves Labs | GFP-1020 | (1:3000) |
| Antibody | (Mouse monoclonal) anti-Sox2 | R&D Systems | AF2018 | (1:500) |
| Antibody | (Rabbit polyclonal) anti-Opn1mw | Millipore Sigma | AB5405 | (1:500) |
| Antibody | (Goat polyclonal) anti-Aldh1a1 | Abcam | 9,883 | (1:1000) |
| Sequence-based reagent | Mm *Arr3* RNAscope Probe | Advanced Cell Diagnostics | 486,551 | |
| Sequence-based reagent | Mm *Nrl* RNAscope Probe | Advanced Cell Diagnostics | 475,011 | |

## Mouse

All animals were handled according to protocols approved by the Institutional Animal Care and Use Committee (IACUC) of Harvard University (IACUC protocol: 1695). RARE-LacZ mice (stock #008477), FVB/rd1 mice (stock #207), rd10 mice (stock #004297), CAG-CreER mice (stock #004682), and Pdgfra-Cre mice (stock #013148) were obtained from Jackson Laboratory. *Rho*$^{-/-}$ mice were a gift from Janis Lem, Tufts University, Boston, MA (*Lem et al., 1999*). Aldh1a1, 2, 3 flox/flox mice were a gift from Norbert Ghyselinck, IGBMC, France. All retina degeneration mouse lines were housed within ±2 rows in their respective racks to control for light exposure. For tissue harvest, mice were euthanized with $CO_2$ and then secondarily with cervical dislocation. All genotyping primer sequences are in Supplementary Files.

## Human retina samples

Frozen eyes were obtained from Restore Life USA (Elizabethton, TN) through TissueForResearch. Patient DRLU031318A was a 47-year-old female with no clinical eye diagnosis and the postmortem interval was 6 hr. Patient DRLU032618A was a 52-year-old female with no clinical eye diagnosis and the postmortem interval was 8 hr. Patient DRLU041518A was a 57-year-old male with no clinical eye diagnosis and the postmortem interval was 5 hr. Patient DRLU041818C was a 53-year-old female with no clinical eye diagnosis and the postmortem interval was 9 hr. Patient DRLU051918A was a 43-year-old female with no clinical eye diagnosis and the postmortem interval was 5 hr. This IRB protocol (IRB17-1781) was determined to be not human subjects research by the Harvard University-Area Committee on the Use of Human Subjects.

## Plasmids

Rho-Cre and Cralbp-Cre were from *Matsuda and Cepko, 2007*. Newly generated plasmids (CAG-RARa-VP16, CALNL-RARa-VP16, TRE-RARa-VP16, CAG-rtTA, AAV-CMV-Cyp26a1) have been deposited to Addgene.

## Retina flatmount IHC

Eyes were enucleated, and the retinas (with the lens) were dissected in PBS. These retinas were fixed in 4% PFA (Electron Microscopy Sciences, cat. #15714S, diluted in PBS) for 20 min at RT. After 2 x

washes with PBS, they were incubated in Blocking Buffer (0.3% Bovine Serum Albumin (Jackson ImmunoResearch, cat. #001-000-162), 4% donkey serum (Jackson ImmunoResearch, cat. #017-000-121), 0.3% Triton X-100 (Sigma Millipore, cat. #T8787), diluted in PBS) for 15 min at RT. Retinas were incubated in primary antibody, diluted in Blocking Buffer, for 2–4 hr at RT with the following antibodies at the following concentrations: rabbit Cone Arrestin (1:3000, Millipore Sigma, cat. #AB15282), chicken B-galactosidase (1:3000, Aves Labs, cat. #BGL1010), goat B-galactosidase (1:3000, AbD Serotec, cat. #4600–1409), mouse Brn3a (1:500, Millipore Sigma, cat. #MAB1585), rabbit Iba1 (1:500, GeneTex, cat. #GTX100042), mouse Glutamine Synthetase (1:3000, Millipore Sigma, cat. #MAB302), goat Choline Acetyltransferase (1:500, Millipore Sigma, cat. #AB144P), chicken GFP (1:3000, Aves Labs, cat. #GFP-1020), mouse Sox2 (1:500, R&D Systems, cat. #AF2018), rabbit Opn1mw (1:500, Millipore Sigma, cat. #AB5405), goat Aldh1a1 (1:1000, Abcam, cat. #9883, 0.5% Triton X-100 was used in the Blocking Buffer for this antibody). After 3 x wash with PBS, the retinas were incubated in corresponding secondary antibodies conjugated to AlexaFluor 488, 594, or 647, diluted in Blocking Buffer (1:750, Jackson ImmunoResearch) for 1 hr at RT. After 3 x washes with PBS, the lens were removed, and the retinas were flatmounted on a No. 1.5 coverslip (VWR, cat. #48393–241) and dried. The coverslips were mounted on a Superfrost Plus slides (Fisher Scientific, cat. #12-550-15) with Fluoromount-G (SouthernBiotech cat. #0100–01) and dried overnight at RT.

## Retinal flatmount LacZ staining

LacZ staining on RARE-LacZ flatmount was performed as described previously (*da Silva and Cepko, 2017*).

## Retina Cryosectioning

Eyes were enucleated, and the retinas (with the lens) were dissected in PBS. The retinas were fixed in 4% PFA for 20 min at RT. For smFISH experiments, it was critical to use freshly opened PFA ampule for fixation. After 2 x washes with PBS, the lens were removed. For experiments that required only the electroporated region, the BFP$^+$ area was dissected out under a fluorescent dissection scope. The retina was incubated in 30% sucrose/PBS (Millipore Sigma, cat. #S0389) until the tissue sank to the bottom. The tissue was embedded in 50%/15% OCT/Sucrose solution (100% OCT mixed with 30% sucrose, equal parts, Tissue-Tek, cat. #25608–930) in a cryomold. The retinas were snap-frozen in dry ice/ethanol slurry and kept at –80 °C. The retinas were cryosectioned at 30 µm thickness and mounted onto Superfrost plus slides. The slides were stored at –80 °C.

## Retina section IHC

Slides with retina sections were 2 x washed with ~3 mL of PBS and then dried completely. The slides were washed 1 x with PBS. They were incubated with 500 µL of Blocking Buffer for 15 min at RT and then with 500 µL of primary antibody diluted in Blocking Buffer for 2 hr at RT. After 3 x washes with PBS, the slides were incubated with 500 µL of secondary antibody diluted in Blocking Buffer for 1 hr at RT. After 3 x washes with PBS, the slides were dried completely and coverslipped with Fluoromount-G.

## Single molecule FISH

SABER FISH and RNAscope were used for smFISH of retinal sections (*Kishi et al., 2019*). For SABER, the slides were washed with PBS for 5–10 min to remove the OCT on the slides. Subsequently, sections were completely dried, and an adhesive chamber (Grace Bio-Labs, cat. #621502) was placed to encase the sections. The samples were incubated in 0.1% PBS/Tween-20 (MilliporeSigma, cat. #P9416) for 10 min. The PBST was removed, and the samples were incubated with pre-warmed (43 °C) 40% wHyb (2 x SSC (Thermo Fisher Scientific, cat. #15557044), 40% deionized formamide (MilliporeSigma, cat. #S4117), 1% Tween-20, diluted in UltraPure Water) for at least 15 min at 43 °C. The 40% wHyb was removed, and the samples were incubated with 100 µL of pre-warmed (43 °C) Probe Mix (1 µg of probe per gene, 96 µL of Hyb1 solution (2.5 x SSC, 50% deionized formamide, 12.5% Dextran Sulfate (MilliporeSigma cat. #D8906), 1.25% Tween-20), diluted up to 120 µL with UltraPure Water) and incubated 16–48 hr at 43 °C. The samples were washed twice with 40% wHyb (30 min/wash, 43 °C), twice with 2 x SSC (15 min/wash, 43 °C), and twice with 0.1% PBST (5 min/wash, 37 °C). The samples were then incubated with 100 µL of Fluorescent Oligonucleotide Mix (100 µL of PBST, 2 µL of each 10 µM Fluorescent Oligonucleotide) for 15 min at 37 °C. The samples were washed three times with PBST at

37 °C for 5 min each and counterstained with DAPI (Thermo Fisher Scientific, cat. #D1306; 1:50,000 of 5 mg/mL stock solution in PBS). The oligo sequences for the probes (*Aqp4, Aldh1a1, Cyp26a1*) are in Supplementary Files.

For RNAscope, the RNAscope Fluorescent Multiplex Assay was used with 30 µm cryosections according to protocol (Advanced Cell Diagnostics). The following probes were used: Mm-Arr3 (cat. #486551), Mm-Nrl (cat. #475011).

## In vivo electroporation

Electroporation of DNA plasmids into neonatal mouse retina was performed as described previously (*Matsuda and Cepko, 2004*; *Wang et al., 2014*). Briefly, glass needles were created by pulling Wiretrol II capillaries (Drummond Scientific Company, cat. #5-000-2005) using a needle puller (Sutter Instrument, Model P-97). The glass needles were beveled on two edges with a microgrinder (Narishige, cat. #EG-401). 10 µL of DNA plasmids (1 µg/µL per construct) was prepared with 0.5 µL of 2.5% FastGreen (Millipore Sigma cat. #F7252). P0 – P1 mouse pups were anesthetized by cryoanesthesia on ice. A small skin incision was made by a 30-gauge needle and the eye was exposed. The beveled glass needle containing the DNA plasmid was inserted through the sclera into the subretinal space, and the DNA solution was injected into the ventral hemisphere using a Femtojet Express pressure injector (Eppendorf, cat. #920010521) with the following setting: 330 Pa, 3 s. The eyelids were then closed with a cotton applicator. Tweezer-type electrodes (Harvard Apparatus, BTX, model 520, 7 mm diameter, cat. #450165) were placed with the positive end slightly above the injected eyelid. An electric field was applied using an electroporator (BEX, cat. #CUY21EDIT) with the following parameters: Volts: 80 V, Pulse-On: 50ms, Pulse-Off: 950ms, Number of pulses: 5. Then, the eyelids were dried with a cotton applicator to ensure eyelid closure.

## AAV production

HEK293T cells were seeded onto five 15 cm plates (Celltreat cat. #229651) and grown in 10% FBS/DMEM/PS media (Thermo Fisher Scientific, cat. #10437028, cat. #11995065, cat. #15140163). At 100% confluency, the cells were transfected using 340 µL of polyethylenimine (PolyScience, cat. #24765–2, diluted to 1 mg/mL) with the following DNA plasmids: 35 µg of ShH10Y capsid (a gift from John Flannery, Berkeley, CA), 35 µg of AAV-CMV-Cyp26a1 or AAV-CMV-GFP vector, 100 µg of pHGTI-adeno1. The cells were incubated in 10% NuSerum/DMEM/PS (BD Biosciences, cat. #355500) for 24 hr. Then, the media was changed to DMEM without serum for 48 hr. Next, the cells and media were collected and centrifuged at 1000 xg for 5 min at RT. The cell pellet (1) and the media (2) were separated for downstream processing. (1) The cell pellet was resuspended in 10 mL of lysis buffer (150 mM NaCl, 20 mM Tris-HCl pH 8.0) and underwent 3 x freeze-thaw between dry ice/ethanol bath and 37°C water bath. 10 µL of 1 M MgCl2 and 10 µL of Benzonase (Millipore Sigma, cat. #E1014) were added and incubated at 37 °C for 15 min. It was then centrifuged at 3,800 xg for 20 min at 4 °C, and the supernatant was collected. (2) The media was filtered through a 0.45 µm CA filter (Corning, cat. #430768). NaCl (to 0.4 M) and PEG8000 (to 8.5% w/v, Sigma, cat. #81268) were added over a period of 1.5 hr with stirring at 4 °C. It was then centrifuged at 7,000 xg for 10 min at 4 °C and the supernatant was discarded. The supernatant from (1) was added and the viral pellet was resuspended. This solution was overlaid on top of a gradient (17%, 25%, 40%, 60%) of Iodixanol (Millipore Sigma, cat. #D1556) and ultracentrifuged at 46,500 rpm (Beckman Coulter VTi50) for 1.5 hr at 16 °C. The viral fraction in the 40% gradient was harvested with a 21-gauge needle, and PBS was added up to 15 mL. The solution was transferred to an Amicon tube with a centrifugal filter (Millpore Sigma, cat. #UFC910024) and centrifuged at 1,500 xg for 15 min at 4 °C. This process was repeated three times with fresh PBS. At the end of the PBS washes, the remaining viral volume (~100 µL) was collected and titered based on the intensity of VP1, VP2, and VP3 proteins on an SDS-PAGE gel.

## AAV injection

All AAV injections were performed at P4. Subretinal injections were performed as described above (under in vivo Electroporation). Per eye, approximately 5E8 vector genomes (vg) per AAV were delivered into the dorsal subretinal space.

## Doxycycline administration

At P20, mice electroporated with or without CAG-rtTA and TRE-RARa-VP16 constructs were fed Doxycycline-added diet (2,000 mg/kg, Envigo, cat. #TD.05512) ad libitum. The diet was stored at 4 °C and replaced once per week.

## Tamoxifen injection

Aldh Flox mice were intraperitoneally injected with tamoxifen, diluted in corn oil (20 mg/mL, 100 μL per pup, Millipore Sigma, cat. #T5648) at P10. To dissolve the tamoxifen in corn oil (Millipore Sigma, cat. #C8267), it was heated to 65 °C in a water bath for 1–2 hr with occasional vortexing. The dissolved tamoxifen was aliquoted and stored at 4 °C for less than 1 week.

## Statistical analysis

Normality was tested using the Shapiro-Wilk test. For parametric datasets, Student's two-tailed T test, one-way ANOVA with Tukey's multiple comparison test, and two-way ANOVA were performed to compare between control and experimental groups. For nonparametric datasets, Mann-Whitney test was performed. Multiple litters were used for each experiment, but each litter contained both control and experimental groups for all experiments to account for litter-to-litter variability. Both sexes were used randomly.

## ddPCR

Retina or liver tissue was harvested into 500 μL of Trizol (Thermo Fisher Scientific, cat. #15596026). The tissue was homogenized with disposable pestles (Fisher Scientific, cat. #12141368), and RNA was extracted according to the Trizol protocol. cDNA was generated using either the Superscript 3 or 4 First Strand Synthesis System (Thermo Fisher Scientific, cat. #18080051 or #18091050). For ddPCR, QX200 EvaGreen 2 x Supermix was used (BioRad, cat. #1864034) with 0.5 μm of primers and appropriate amounts of cDNA. The primer sequences can be found in Supplementary Files.

## Cell-type-specific bulk RNA sequencing

Neural retinas (without RPE) from P28 – P36 CD1 WT mice were dissected in PBS and flatmounted. The peripheral 1/3 of each flatmount petal, each comprising roughly one quarter of each retina, was collected as peripheral retina tissue. The central 1/3 was collected as central retina tissue. The mid-peripheral 1/3 was discarded. The retina was then transferred to a microcentrifuge tube and incubated for 7 min at 37 °C with an activated papain dissociation solution (87.5 mM HEPES pH 7.0 (Thermo Fisher Scientific, cat. #15630080), 2.5 mM L-Cysteine (MilliporeSigma, cat. # 168149), 0.5 mM EDTA pH 8.0 (Thermo Fisher Scientific, cat. #AM9260G), 10 μL Papain Suspension (Worthington, cat. #LS0003126), 19.6 μL UltraPure Nuclease-Free Water (Thermo Fisher Scientific, cat. #10977023), HBSS up to 400 μL, activated by a 15 min incubation at 37 °C). The retina was then centrifuged at 600 xg for 3 min. The supernatant was removed, and 1 mL of HBSS/10% FBS (Thermo Fisher Scientific, cat. #10437028) was added. The pellet was then centrifuged at 600 xg for 3 min. The supernatant was removed, and 600 μL of trituration buffer (DMEM (Thermo Fisher Scientific, cat. #11995065), 0.4% (wt/vol) Bovine Serum Albumin (MilliporeSigma cat. #A9418)) was added. The pellet was dissociated by trituration at room temperature (RT) using a P1000 pipette up to 20 times or until the solution was homogenous. For all remaining solutions, 5 μL mL$^{-1}$ RNasin Plus (Promega, cat. #N2611) was added 10 min before use. The cells were centrifuged at 600 xg for 3 min, and the pellet was resuspended in 1 mL of 4% PFA and 0.1% saponin (Millipore Sigma, cat. #47036) for 30 min at 4 °C. After 2 x wash in Wash Buffer (0.1% saponin, 0.2% BSA), the cells were incubated in primary antibody (Cone Arrestin, 1:1000, in 0.1% saponin/1% BSA) for 30 min at 4 °C. The cells were washed twice in Wash Buffer and incubated in secondary antibody (Donkey anti-rabbit 647 in 0.1% saponin/1% BSA) for 30 min at 4 °C. After 1 x wash, the sample was passed through a 35 μm filter (Thermo Fisher Scientific, cat. #352235) before proceeding to FACS. FACSAria (BD Biosciences) with 488, 561, 594, and 633 nm lasers was used for FACS. CAR$^+$ population was sorted into microcentrifuge tubes with 500 μL of PBS and kept on ice after FACS. The sorted cells were transferred to a 5 mL polypropylene tube and centrifuged at 3000 xg for 7 min at RT. The supernatant was removed, and the cells were resuspended in 100 μL Digestion Mix (RecoverAll Total Nuclear Isolation Kit (Thermo Fisher Scientific, cat. #AM1975) 100 μL of Digestion Buffer, 4 μL of protease), and incubated for 3 hr at 50 °C, which differs from

the manufacturer's protocol. The libraries for RNA sequencing were generated using SMART-Seq v.4 Ultra Low Input RNA (Takara Bio, cat. #634890) and Nextera XT DNA Library Prep Kit (Illumina, cat. #FC1311096) according to the manufacturer's protocol. 11 cycles were used for the SMART-Seq. The cDNA library fragment size was determined by the BioAnalyzer 2100 HS DNA Assay (Agilent, cat. #50674626). The libraries were sequenced as 75 bp paired-end reads on NextSeq 500 (Illumina). The RNA-seq data was analyzed as described previously (*Amamoto et al., 2019*).

## Image acquisition

Fluorescent flatmount images were acquired with Nikon Ti inverted widefield microscope with a Prior ProScanIII motorized stage. The objective used was Plan Apo Lambda 10 x/0.45 Air DIC N1 objective, and the camera used was Hamamatsu ORCA-Flash 4.0 V3 Digital CMOS camera. Fluorescent retina section images were acquired with W1 Yokogawa Spinning disk confocal microscope with 50 μm pinhole disk and 488, 561, and 640 laser lines. The objectives used were either Plan Apo 20 x/0.75 air or Plan Apo 60 x/1.4 oil objectives, and the camera used was Andor Zyla 4.2 Plus sCMOS mono-chrome camera. Nikon Elements Acquisition Software (AR 5.02) was used for image acquisition and Fiji or Adobe Photoshop CS6 was used for image analysis.

## Cone quantification

Cones were quantified on widefield images at the plane of the cone cell bodies. For electroporated retinas, the number of cones were quantified in central BFP$^+$ regions as schematized in *Figure 3—figure supplement 2*. Briefly, a circle of 1.5 mm in radius was drawn with the center overlaid on the optic nerve head. This method was used to ensure that the peripheral cones were not quantified. Within this circle, the BFP$^+$ region was used for counting CAR$^+$ cones. For unelectroporated retinas, a 640 μm x 640 μm square box was drawn in the middle of the dorsal region, 70% of the length from the optic nerve head, as schematized in *Figure 2D*. For all quantifications, cones were counted blinded using the Cell Counter function in Fiji, and in some instances, verified by a second blinded counter. For all comparisons between conditions, both conditions were included among littermates to account for litter-to-litter variability. The area of electroporated region was calculated using Fiji.

## Acknowledgements

We thank former and current members of the Cepko and Tabin Labs for the insightful discussions and feedback. We thank PM Llopis, R Stephansky, and the MicRoN core at Harvard Medical School for their assistance with microscopy. We thank C Araneo, F Lopez, and the Flow Cytometry Core Facility for their assistance with flow cytometry. We thank J Patrice and the HCCM animal facility for their assistance with mouse husbandry. This work was supported by the Howard Hughes Medical Institute (CLC), NIH K99/R00 Pathway to Independence Award (RA K99EY032110), and Edward R and Anne G Lefler Postdoctoral Fellowship (RA).

## Additional information

### Funding

| Funder | Grant reference number | Author |
| --- | --- | --- |
| Howard Hughes Medical Institute | | Constance L Cepko |
| National Eye Institute | K99EY032110 | Ryoji Amamoto |
| Edward R. and Anne G. Lefler Postdoctoral Fellowship | | Ryoji Amamoto |

The funders had no role in study design, data collection and interpretation, or the decision to submit the work for publication.

## Author contributions
Ryoji Amamoto, Conceptualization, Data curation, Formal analysis, Funding acquisition, Investigation, Methodology, Project administration, Resources, Software, Supervision, Validation, Visualization, Writing - original draft, Writing - review and editing; Grace K Wallick, Data curation, Investigation, Validation; Constance L Cepko, Conceptualization, Funding acquisition, Project administration, Resources, Supervision, Writing - review and editing

## Author ORCIDs
Ryoji Amamoto (ID) http://orcid.org/0000-0002-9335-112X
Constance L Cepko (ID) http://orcid.org/0000-0002-9945-6387

## Ethics
Human subjects: This IRB protocol (IRB17-1781) was determined to be not human subjects research by the Harvard University-Area Committee on the Use of Human Subjects.
All animals were handled according to protocols approved by the Institutional Animal Care and Use Committee (IACUC) of Harvard University (IACUC protocol: 1695).

## Decision letter and Author response
Decision letter https://doi.org/10.7554/eLife.76389.sa1
Author response https://doi.org/10.7554/eLife.76389.sa2

---

# Additional files

## Supplementary files
• Supplementary file 1. Genotyping and ddPCR primers and oligonucleotide sequences for SABER FISH probes.
• Transparent reporting form

## Data availability
Raw sequencing data and matrices of read counts are available at GEO:GSE186612.

The following dataset was generated:

| Author(s) | Year | Dataset title | Dataset URL | Database and Identifier |
|---|---|---|---|---|
| Amamoto R, Cepko C | 2021 | Retinoic acid signaling mediates peripheral cone photoreceptor survival in a mouse model of retina degeneration | https://www.ncbi.nlm.nih.gov/geo/query/acc.cgi?acc=GSE186612 | NCBI Gene Expression Omnibus, GSE186612 |

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
