## [Editor Report]

Retinitis pigmentosa (RP) is a heterogeneous condition that leads to photoreceptor cell death and thus to different degree of blindness. The degeneration is often caused by mutations in genes expressed in rods but cones end up degenerating with a mutation-independent process. Thus, identifying how to save cones would increase the quality of life of RP patients, bypassing the tremendous genetic heterogeneity of the disease. This study shows that cones positioned in the periphery of the mouse retina is more resistant to cell death and investigates the reasons of this resilience. Using different genetic approaches, the authors elegantly show that retinoic acid signaling derived from Muller glial cells located in the periphery of the mouse retina is implicated in local survival of cone photoreceptors in mouse models of RP. This center-to-periphery asymmetry is also present in the human retina although the clinical significance of the observation remains to be tested. This well executed study will be of high interest to vision and ophthalmology science community

---

## [Decision Letter]

**Decision letter after peer review:**

Thank you for submitting your article "Retinoic acid signaling mediates peripheral cone photoreceptor survival in a mouse model of retina degeneration" for consideration by *eLife*. Your article has been reviewed by 3 peer reviewers, one of whom is a member of our Board of Reviewing Editors, and the evaluation has been overseen by Tirin Moore as the Senior Editor. The following individual involved in review of your submission has agreed to reveal their identity: Henri Leinonen (Reviewer #3).

Essential revisions:

1) Please revise statistical analysis and provide details on how normal distribution was tested.

2) In Figure 4, A and B are redundant. Please eliminate panel B. Furthermore, the RNA-seq comparison obtained in mouse could be compared with available human retinal transcriptomic database, including comparisons of human peripheral vs. central retina.

3) Sentences such as that RA signaling is necessary and sufficient should be toned down as a number of cones seems to survive even in absence of RA.

4) The significance of the findings for humans should be discussed in view of the notion that in most cases of human RP degeneration starts in the periphery and progressed to the center.

*Reviewer #1 (Recommendations for the authors):*

This is well conducted and interesting study however the conclusions are overstated.

1) Stating that RA signaling is necessary and sufficient for peripheral cone survival seems inaccurate. Figure 2 shows that after inactivation of the different RA synthetizing enzymes a considerable number of cones is still present in the retinal periphery. Thus, other factors are likely involved. Have the authors attempted to investigate other factors found in the RNA-seq analysis?

2) Panel A and B of figure 4 are redundant. They are different representations of the same data. Panel B should be removed.

3) The significance of ALDH1A1 expression in the human retinal periphery is unclear. In the large majority of human RP cases, rod degeneration starts in the retinal periphery and patients are left with tunnel vision. This should be discussed in a clear manner. At present the discussion suggests that what found in mouse is relevant in humans but a more realistic view should be provided.

4) The pathogenesis of AMD is different from that of RP, even thus in both there is photoreceptor degeneration. Mixing RP and AMD in the discussion do not seem to be appropriate.

*Reviewer #2 (Recommendations for the authors):*

The study reported here stems from a long term interest in searching for mechanisms responsible for the bystander degeneration of cones in Retinitis Pigmentosa. Cones have fundamental relevance in human vision and therefore the study is timely and important, also being based on the fact that cone rescue is achieved using mutation independent approaches, capable of bypassing the tremendous genetic heterogeneity of the disease.

The primary observation of the study is that in mouse models of RP, cones in the far peripheral retina survive long despite complete rod loss. The Authors succeed in demonstrating that a fundamental mechanism responsible for this survival implicates active retinoic acid (RA) signaling in peripheral Muller glia, which turns out to be both sufficient and necessary for the extended cone survival. Mice deficient in synthesis of RA in a retinal degeneration background were constructed, demonstrating that in these conditional knockout of RA an abrogation of peripheral cone survival was observed. Conversely, constitutive activation of RA signaling in the central retina promoted long-term cone survival. Human retinal samples used to reveal RA signaling molecules by ddPCR confirmed an asymmetric expression of this molecular system, which predominates in the peripheral retina, similar to what found in mice.

Hence, the data indicate that RA is one molecule whose signaling can enhance cone survival, providing a mutation independent tool to protect these cells from bystander degeneration.

The study is supported by a logical and excellently performed sequence of experiments based on highly modern methods, and comprising genetics, molecular biology, and microscopy tools. The text is clearly written and adequately illustrated. The study is important and the data of highly potential value from a translational point of view.

Few suggestions are provided below:

line 109: please specify better the fact that peripheral cone survival occurs in the total absence of rods and it is not secondary to a higher rate of rod survival, contributing to cone viability.

line 51: Among the causes of bystander effect, local inflammation should

be mentioned (i.e. Guadagni et al., Faseb J 2012; Karlstetter M, PRER 2015);

line 185. In Figure 1.4, some overlapping of the staining with the Muller cell marker is evident but only partially. Can one really say that MOST of the two signals really overlap? In addition, no processes of Muller cells are labelled in in the inner part of the retina. Is there an explanation?

line 739: were cones counted on z-stacks images or at the cell body focal plane? Please specify. Can you give an example of the average number of cones counted per field? This gives an idea of the sampling error for each retinal specimen.

Do cones that survive in virtue of the activation of RA signaling retain outer segments or not? This would indicate higher potential of functional rescue achieved with this manipulation. The Authors might want to discuss this point.

Is there an advantage or an adaptive effect in concentrating RA sensitivity to the peripheral retina? Any explanation or speculation the Authors can add to the Discussion?

*Reviewer #3 (Recommendations for the authors):*

Suggestions:

– In line 156, why do you think observed amount of CAR+ cells is 5-times higher than expected?

– In one supplementary Figure legend you are talking about Cd1 mice. Is this correct, or should it be Rd1 mouse?

– Student t-test (parametric, required normal equal variances between the groups if not corrected) was used as general statistical tests. Is it correct in Figure 2B, 2G, 3H, 3J? Rather, should you not use non-parametric Mann-Whitney test?

– Similar critique for 3I. I am not sure if ANOVA is correct as variances between the groups do not seem equal. Did you run a test for this? Perhaps 3I should be analyzed with Kruskal-Wallis.

– 4A: How can you run 2-way ANOVA here? Again, Mann-Whitney would be fine and will give you P<0.01 with that sample size.

– With respect to the human samples, how did you choose to test only one target gene? Why not to do a bit more comprehensive investigation since you have the samples? Rhd10, other Aldh subforms? RNA-sequencing?

– Continuing the previous. Why not use already published retinal transcriptomics data from post mortem human samples, freely accessible through GEO?

– In rows 469-471, do you mean that CONE degeneration is stronger in central retina in RP patients? Isn´t tunnel vision a hallmark of RP, caused by loss of PERIPHERAL rods initially?

– Retinoic acid activity may be inhibited with pharmacology using disulfiram (Telias…Kramer 2019, Neuron). I believe the genetic tools you used answer the questions you raised but just wondering, did you consider testing the context using pharmacology?

---

## [Author Response]

Essential revisions:1) Please revise statistical analysis and provide details on how normal distribution was tested.

Thank you for the suggestion. We have revised to account for normality of each dataset using the Shapiro-Wilks test. In the Materials and methods section under Statistical Analysis, we have added:

“Normality was tested using the Shapiro-Wilk test. For parametric datasets, Student’s twotailed T test, one-way ANOVA with Tukey’s multiple comparison test, and two-way ANOVA were performed to compare between control and experimental groups. For nonparametric datasets, Mann-Whitney test was performed.”

Two datasets were not normally distributed – Figures 2K and 3E. For these datasets, we used the Mann-Whitney test, and the differences remained significant.

2) In Figure 4, A and B are redundant. Please eliminate panel B. Furthermore, the RNA-seq comparison obtained in mouse could be compared with available human retinal transcriptomic database, including comparisons of human peripheral vs. central retina.

We have removed Figure 4B. We have also added in the Results:

“We obtained five postmortem adult human non-RP retinas to investigate the expression level of *ALDH1A1* in the central and peripheral retina. As assessed by ddPCR, *ALDH1A1* was significantly enriched in the peripheral retina (Figure 4). Furthermore, we compared our RNA-seq dataset with previously-published human transcriptional profiling of central and peripheral retina. In addition to the peripherally-enriched *ALDH1A1*, *PRSS56* was enriched in the peripheral human retina while *SNCG* was enriched in the central human retina^61,62^. This result opens the possibility that RA signaling is active in the human peripheral retina and may play a role in preserving peripheral vision, if any. In addition, other pathways which differ between peripheral and central retina may be conserved and might deserve investigation.”

3) Sentences such as that RA signaling is necessary and sufficient should be toned down as a number of cones seems to survive even in absence of RA.

In the Results section, we added:

“Accordingly, quantification of CAR^+^ cones in sampled dorsal regions showed a significant reduction in the number of cones (Mean ± SD: -CreER: 40.94±13.51; +CreER: 23.9±12.74), suggesting that RA is necessary for the locally enriched peripheral cone survival (Figure 2E). However, despite the near complete depletion of Aldh1a1 from the retina, some cones persisted in the dorsal periphery. Therefore, it is possible that other intrinsic or extrinsic pathways contribute to promote peripheral cone survival.”

4) The significance of the findings for humans should be discussed in view of the notion that in most cases of human RP degeneration starts in the periphery and progressed to the center.

We have asked multiple ophthalmologists about this issue, and have learned the following. They summarize RP as peripheral to central vision loss. However, they typically do not image the far periphery using OCT, or assay the far periphery using vision tests. Most studies do not go out more than 90 degrees, whereas we are examining tissue in the mouse that is >100 degrees. It is thus unclear whether there is preservation of far peripheral cones. However, the statement of peripheral to central is not accurate if one examines the literature, as it is mid-peripheral to central.

We have added the following to the Results section to address this issue:

“Whether RA signaling plays a role in preserving cones in human RP patients is not known. For a typical patient with RP, photoreceptor degeneration starts in the mid-peripheral retina, leading to loss of visual function in that region^29^. Degeneration progresses over time towards the central retina, including the macula, with subsequent loss of central vision. However, peripheral vision is typically not assayed in humans, though islands of vision have been observed in some RP patients by functional visual testing^22,60^. A systematic study of the geography of vision loss, which includes the far periphery, and the physical correlates, has not been done, but might yield insights into whether and how such islands may be preserved in humans.”

Reviewer #1 (Recommendations for the authors):This is well conducted and interesting study however the conclusions are overstated.1) Stating that RA signaling is necessary and sufficient for peripheral cone survival seems inaccurate. Figure 2 shows that after inactivation of the different RA synthetizing enzymes a considerable number of cones is still present in the retinal periphery. Thus, other factors are likely involved. Have the authors attempted to investigate other factors found in the RNA-seq analysis?

See above under Essential Revisions.

2) Panel A and B of figure 4 are redundant. They are different representations of the same data. Panel B should be removed.

See above under Essential Revisions.

3) The significance of ALDH1A1 expression in the human retinal periphery is unclear. In the large majority of human RP cases, rod degeneration starts in the retinal periphery and patients are left with tunnel vision. This should be discussed in a clear manner. At present the discussion suggests that what found in mouse is relevant in humans but a more realistic view should be provided.

See above under Essential Revisions.

4) The pathogenesis of AMD is different from that of RP, even thus in both there is photoreceptor degeneration. Mixing RP and AMD in the discussion do not seem to be appropriate.

While the molecular mechanisms underlying photoreceptor degeneration in AMD and RP are likely different, both diseases exhibit increased susceptibility in the central retina, which expresses Cyp26a1. Therefore, we believe that a discussion of spatial pattern of retinal degeneration in these diseases is appropriate for our manuscript.

Reviewer #2 (Recommendations for the authors):The study reported here stems from a long term interest in searching for mechanisms responsible for the bystander degeneration of cones in Retinitis Pigmentosa. Cones have fundamental relevance in human vision and therefore the study is timely and important, also being based on the fact that cone rescue is achieved using mutation independent approaches, capable of bypassing the tremendous genetic heterogeneity of the disease.The primary observation of the study is that in mouse models of RP, cones in the far peripheral retina survive long despite complete rod loss. The Authors succeed in demonstrating that a fundamental mechanism responsible for this survival implicates active retinoic acid (RA) signaling in peripheral Muller glia, which turns out to be both sufficient and necessary for the extended cone survival. Mice deficient in synthesis of RA in a retinal degeneration background were constructed, demonstrating that in these conditional knockout of RA an abrogation of peripheral cone survival was observed. Conversely, constitutive activation of RA signaling in the central retina promoted long-term cone survival. Human retinal samples used to reveal RA signaling molecules by ddPCR confirmed an asymmetric expression of this molecular system, which predominates in the peripheral retina, similar to what found in mice.Hence, the data indicate that RA is one molecule whose signaling can enhance cone survival, providing a mutation independent tool to protect these cells from bystander degeneration.The study is supported by a logical and excellently performed sequence of experiments based on highly modern methods, and comprising genetics, molecular biology, and microscopy tools. The text is clearly written and adequately illustrated. The study is important and the data of highly potential value from a translational point of view.Few suggestions are provided below:line 109: please specify better the fact that peripheral cone survival occurs in the total absence of rods and it is not secondary to a higher rate of rod survival, contributing to cone viability.

We have addressed this point empirically by performing FISH against *Nrl* in degenerating rd1 retinas (as explained in the paragraph starting from line 135 in the original manuscript). We believe that this experiment sufficiently addresses this concern. In addition to this statement in the Results section, we have added a line to the Discussion:

“In many mouse models of RP, cones in the far periphery survive long-term despite complete rod degeneration.”

line 51: Among the causes of bystander effect, local inflammation shouldbe mentioned (i.e. Guadagni et al., Faseb J 2012; Karlstetter M, PRER 2015);

These citations have been added.

line 185. In Figure 1.4, some overlapping of the staining with the Muller cell marker is evident but only partially. Can one really say that MOST of the two signals really overlap? In addition, no processes of Muller cells are labelled in in the inner part of the retina. Is there an explanation?

We have increased the contrast of the B-gal staining so that the Muller glial processes are more prominent. While it’s clear that not all Muller glia express B-gal, most of the B-gal^+^ cells are Muller glia. In the Results, we added:

“Accordingly, B-gal staining overlapped with that of Glutamine Synthetase, an MG marker, indicating that most cells with activated RA signaling were MG, although not all peripheral MG were B-gal^+^.”

line 739: were cones counted on z-stacks images or at the cell body focal plane? Please specify. Can you give an example of the average number of cones counted per field? This gives an idea of the sampling error for each retinal specimen.

Cones were quantified from widefield images, not confocal, at the plane of the cell bodies. We added in the Materials and methods:

“Cones were quantified using widefield images at the plane of the cone cell bodies.”

Do cones that survive in virtue of the activation of RA signaling retain outer segments or not? This would indicate higher potential of functional rescue achieved with this manipulation. The Authors might want to discuss this point.

We did not detect an overt preservation of the outer segment structure. However, our previous papers demonstrated functional recovery, as assessed by optomotor, even without rescue of morphologically-intact outer segments.

Is there an advantage or an adaptive effect in concentrating RA sensitivity to the peripheral retina? Any explanation or speculation the Authors can add to the Discussion?

As a speculation, we believe that the peripheral RA signaling is a remnant of its peripheral expression during development. Conditional KO of Aldh1a1, 2, and 3 in wildtype adult retinas did not show overt retinal phenotypes, suggesting that RA signaling does not play a role in maintaining the adult retina. We added some of our thoughts about this in the Ideas and Speculation section.

Reviewer #3 (Recommendations for the authors):Suggestions:– In line 156, why do you think observed amount of CAR+ cells is 5-times higher than expected?

The unexpectedly high number of CAR^+^ cells was likely because some contaminating cells, i.e. Muller glia, which also were labeled by the CAR antibody. This point is demonstrated in Figure 1 —figure supplement 2.

– In one supplementary Figure legend you are talking about Cd1 mice. Is this correct, or should it be Rd1 mouse?

ddPCR in Figure 1 —figure supplement 2 was performed on cells from CD1 mice, not rd1, as was the RNA sequencing experiment in Figure 1.

– Student t-test (parametric, required normal equal variances between the groups if not corrected) was used as general statistical tests. Is it correct in Figure 2B, 2G, 3H, 3J? Rather, should you not use non-parametric Mann-Whitney test?

See above under Essential Revisions.

– Similar critique for 3I. I am not sure if ANOVA is correct as variances between the groups do not seem equal. Did you run a test for this? Perhaps 3I should be analyzed with Kruskal-Wallis.

See above under Essential Revisions.

– 4A: How can you run 2-way ANOVA here? Again, Mann-Whitney would be fine and will give you P<0.01 with that sample size.

See above under Essential Revisions.

– With respect to the human samples, how did you choose to test only one target gene? Why not to do a bit more comprehensive investigation since you have the samples? Rhd10, other Aldh subforms? RNA-sequencing?

While we could have performed RNA sequencing on the human non-RP retina, we believe the published single cell RNA sequencing datasets were sufficient. For our study, we were interested in the spatial expression pattern of Aldh1a1. However, we have added the cross comparison of our dataset with available human RNA-seq datasets, as discussed under Essential Revisions.

– Continuing the previous. Why not use already published retinal transcriptomics data from post mortem human samples, freely accessible through GEO?

See above under Essential Revisions.

– In rows 469-471, do you mean that CONE degeneration is stronger in central retina in RP patients? Isn´t tunnel vision a hallmark of RP, caused by loss of PERIPHERAL rods initially?

See above under Essential Revisions.

– Retinoic acid activity may be inhibited with pharmacology using disulfiram (Telias…Kramer 2019, Neuron). I believe the genetic tools you used answer the questions you raised but just wondering, did you consider testing the context using pharmacology?

We initially tested retinoic acid and disulfiram, but found the negative feedback (i.e. upregulation of Cyp26a1 in response to RA injection) to be troublesome. In addition, such pharmacological administration likely requires daily dosing by IP injection, which we found to increase overall stress and cone degeneration.